# A general deoxygenation approach for synthesis of ketones from aromatic carboxylic acids and alkenes

Muliang Zhang[1], Jin Xie [1] & Chengjian Zhu[1,2]

The construction of an aryl ketone structural unit by means of catalytic carbon–carbon coupling reactions represents the state-of-the-art in organic chemistry. Herein we achieved the direct deoxygenative ketone synthesis in aqueous solution from readily available aromatic carboxylic acids and alkenes, affording structurally diverse ketones in moderate to good yields. Visible-light photoredox catalysis enables the direct deoxygenation of acids as acyl sources with triphenylphosphine and represents a distinct perspective on activation. The synthetic robustness is supported by the late-stage modification of several pharmaceutical compounds and complex molecules. This ketone synthetic strategy is further applied to the synthesis of the drug zolpidem in three steps with 50% total yield and a concise construction of cyclophane-braced 18–20 membered macrocycloketones. It represents not only the advancement for the streamlined synthesis of aromatic ketones from feedstock chemicals, but also a photoredox radical activation mode beyond the redox potential of carboxylic acids.

---

[1] State Key Laboratory of Coordination Chemistry, Jiangsu Key Laboratory of Advanced Organic Materials, School of Chemistry and Chemical Engineering, Nanjing University, Nanjing 210023, China. [2] State Key Laboratory of Organometallic Chemistry, Shanghai Institute of Organic Chemistry, Shanghai 200032, China. Correspondence and requests for materials should be addressed to J.X. (email: xie@nju.edu.cn) or to C.Z. (email: cjzhu@nju.edu.cn)

Aromatic carboxylic acids are extremely promising feedstock chemicals, which can be used to rapidly populate libraries of complex small molecules[1,2]. To date, transition metal-catalyzed decarboxylative coupling enables aromatic carboxylic acids as an alternative source of aryl substructures[3,4]. Examination of thermodynamic data indicates that direct deoxygenative functionalization of aromatic carboxylic acid by activation of C–O bonds remains a serious challenge owing to the similar bond dissociation energies of C–C and C–O bonds (103 vs 102 kcal mol$^{-1}$; Fig. 1a)[5,6].

The concise forging of aryl ketone structural unit by means of carbon–carbon coupling represents the state of the art in synthetic chemistry as they are versatile building blocks for the construction of complex natural products and pharmaceuticals[7]. In order to streamline the synthesis of aryl ketones from aromatic carboxylic acids, certain activating steps are usually necessary to generate acid chlorides[8–12], esters[13–15], and amides[16–20] for subsequent catalyzed or stoichiometric C–C bond formation with nucleophiles, such as organoborons and Grignard reagents (Fig. 1b). Especially, the recent efforts enabled the C–C coupling with mixed anhydride intermediates successful, which can be generated in situ from carboxylic acid and anhydrides[21–23]. However, the requirement of cautious operations and the preformation of intermediates compromises the functional group tolerance and synthetic flexibility together with the increasing demand of late-stage modification of complex target molecules in proteins or living cells under mild conditions. Therefore, the exploration of a practical and sustainable strategy for direct deoxygenative synthesis of ketones in aqueous solution from carboxylic acids is still very desirable but highly challenging.

Although an indirect deoxygenation coupling of acids with a few simple styrenes has been reported via the preformation of reactive anhydride intermediates, a large excess of moisture-sensitive dimethyldicarbonates and (TMS)$_3$SiH reagents were required to initiate the photoredox catalytic cycle[23]. To overcome the mechanistically intrinsic drawbacks, we realized that the diversification of deoxygenation means would potentially offer a conceptually distinct activation mode of carboxylic acids for reaction development. An insight into the classical Wittig reaction[24] inspires us to enquire if the strong P–O affinity between a Ph$_3$P radical cation and a carboxylate anion could facilitate homolytic C–O bond cleavage of carboxylic acids. If feasible, such deoxygenative functionalization of aromatic acids would be independent of the oxidation potential of acids, and thus would significantly expand the synthetic applications. From our continuing efforts in photocatalysis[25–28], we report herein a visible-light-mediated direct deoxygenation activation mechanism of carboxylic acids with cheap triphenylphosphine, which powers deoxygenative C–C coupling of aromatic carboxylic acids with a wide range of alkenes in aqueous solution in the absence of external anhydrides and hydrosilanes additives (Fig. 1c). It affords a general route to aromatic ketones from two easily accessible coupling partners without the involvement of air-sensitive reagents and harsh reaction conditions. This protocol allows practical and friendly reaction conditions which significantly broadens the substrate scope and emphasizes the synthetic application in complex molecules.

## Results

**Reaction optimization.** Initially, the direct deoxygenative C–C coupling of 4-methylbenzoic acid (**1a**) and 2-vinylpyridine (**2a**) was chosen as the model reaction that could be used to optimize the reaction conditions (Table 1 and also Supplementary Table 1). The optimized reaction conditions include 1 mol% of [Ir(dF(CF$_3$) ppy)$_2$(dtbbpy)]PF$_6$ (**I**) as a photocatalyst, 20 mol% K$_2$HPO$_4$ as a base, and 1.2 equiv. Ph$_3$P as an O-transfer reagent in dichloromethane (DCM)/H$_2$O (4:1, v/v) (Table 1, entry 1). Under the standard conditions, the corresponding ketone (**3a**) was obtained in 72% yield. When DCM was employed in place of DCM/H$_2$O, the yield declined significantly, from 72 to 40% (Table 1, entry 2). It was interesting to find that the O-transfer reagent triphenylphosphine was essential for a successful deoxygenative transformation (Table 1, entry 3). Other photocatalysts such as *fac*-Ir (ppy)$_3$ (**II**), Ru(bpy)$_3$(PF$_6$)$_2$ (**III**), eosin Y (**IV**), and Acr$^+$–Mes (**V**) were proved ineffective for this transformation (Table 1, entries 4–7). Control experiments also demonstrated that the reaction could not occur in the absence of either the photocatalyst or light (Table 1, entries 8 and 9).

**Substrate scope of aromatic acids.** With the optimized reaction conditions (Table 1, entry 1) in hand, we investigated the scope of the carboxylic acid substrates (Fig. 2). It was found that acyl radicals generated directly from carboxylic acids could site-selectively add to the β-position of the pyridyl ring with no detectable branched α-position selectivity. In general, aromatic carboxylic acids bearing both electron-donating (e.g., –Me, –Ph, and –BnO) and electron-withdrawing groups (e.g., –F and –Cl) at the *para*-position could react smoothly to produce linear ketones in good yields (**3a–h**). It is noteworthy that 4-bromo (**3g**) and 4-iodo (**3h**) benzoic acids tolerate the conditions well, and this provides an extremely important choice for downstream C–C coupling via palladium catalysis. The substituents on *para*-, *meta*-, and *ortho*-positions of aromatic rings had little influence on the reaction efficiency (**3a–r**). Significantly, carboxylic acids bearing versatile functional groups, such as –NHBoc, –CHO, –COOMe, –OAc, –OH, alkynyl, alkenyl, and acetal are competent reaction partners (**3j–l**, **3n–t**). Terminal alkene and alkyne structural motifs are compatible with this radical transformation (**3q** and **3r**). Heteroaromatic acids, including furan-, thiophene-, quinoline-, and indole-based acids uniformly underwent deoxygenative C–C coupling, furnishing the desired ketones (**3u–x**) in moderate

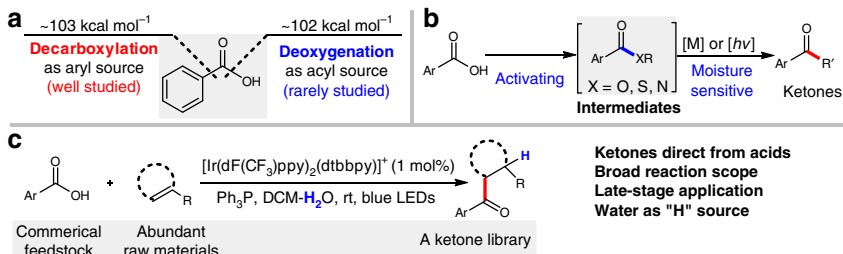

**Fig. 1** Applications of aromatic carboxylic acids in organic synthesis. **a** The nature choice of aromatic carboxylic acid. **b** Known ketone synthesis from acids (indirect strategies). **c** This work: direct deoxygenative C–C coupling by visible light

**Table 1 Optimization of reaction conditions**

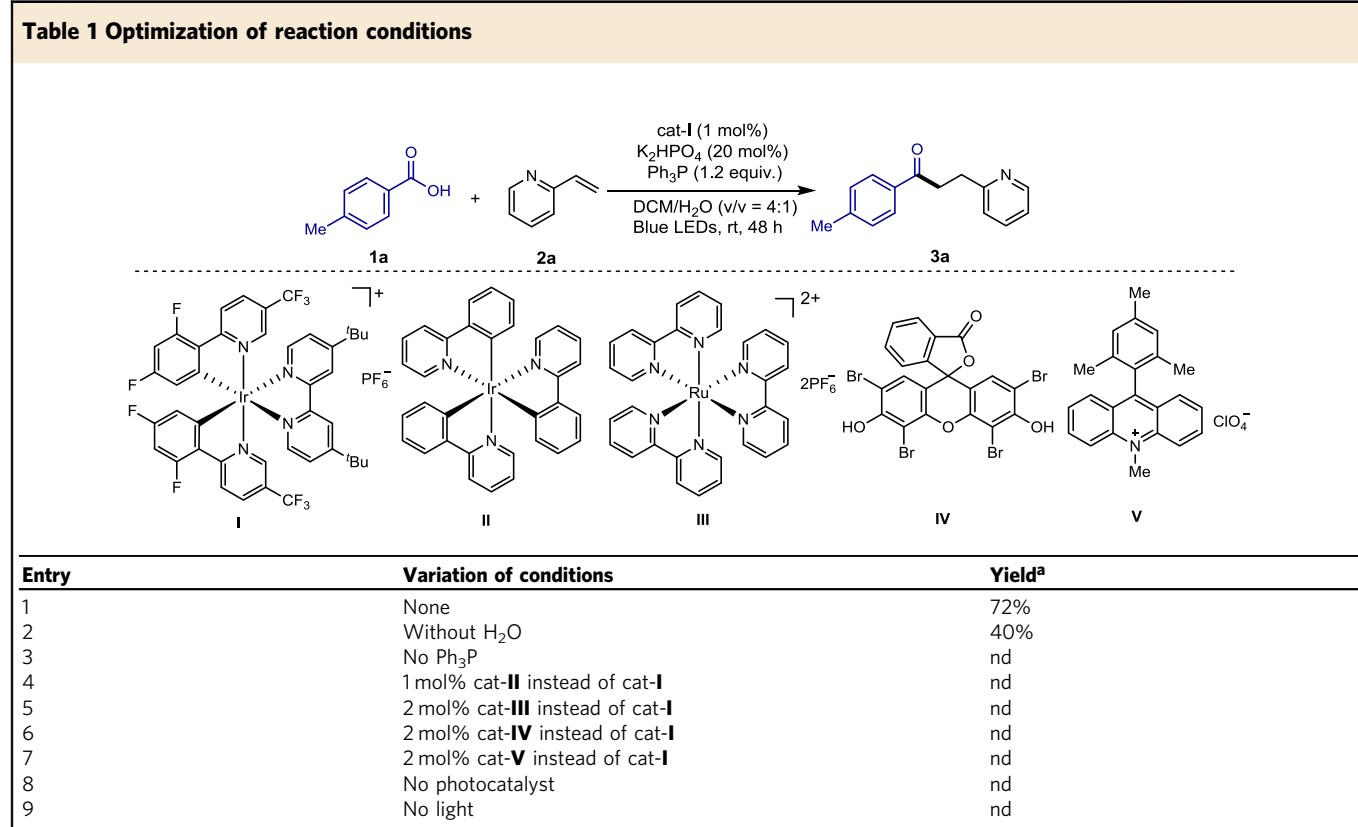

| Entry | Variation of conditions | Yield[a] |
|---|---|---|
| 1 | None | 72% |
| 2 | Without $H_2O$ | 40% |
| 3 | No $Ph_3P$ | nd |
| 4 | 1 mol% cat-**II** instead of cat-**I** | nd |
| 5 | 2 mol% cat-**III** instead of cat-**I** | nd |
| 6 | 2 mol% cat-**IV** instead of cat-**I** | nd |
| 7 | 2 mol% cat-**V** instead of cat-**I** | nd |
| 8 | No photocatalyst | nd |
| 9 | No light | nd |

Standard conditions: photocatalyst **I** (1 mol%), **1a** (0.2 mmol), **2a** (1.5 equiv.), $Ph_3P$ (1.2 equiv.), $K_2HPO_4$ (20 mol%), DCM/$H_2O$ (2.0 mL, v/v = 4:1), rt, 5 W blue light-emitting diodes (LEDs), 48 h
*nd* not detected
[a]Isolated yield

yields. Besides aromatic acids, other kinds of carboxylic acids, including aliphatic carboxylic acids and α,β-unsaturated carboxylic acids however failed in this reaction.

**Substrate scope of alkenes**. Subsequently, many alkenes were examined, giving the results shown in Fig. 3. In view of the prevalence of the pyridine moiety in natural products, pharmaceuticals, and chiral ligands[29,30], a wide range of structurally diverse 2-vinylpyridines possessing different kinds of functional groups were subjected to this protocol. Both electron-rich and electron-poor substituents, including –Me, –Cl, –CHO, –COOEt, –CF₃, –Br, and –OMe at different positions of the pyridyl ring were well tolerated (**3a**, **3y**–**3gg**). The benign compatibility of halogen substituents further emphasized the potential synthetic applications (**3z**, **3ee**, and **3gg**). 4-Vinylpyridines could be converted into the corresponding linear pyridine-based ketones (**3hh**) in 68% yields. Other heteroaromatic styrenes were also efficient substrates (**3ii**–**kk**). When a variety of 1,1-disubstituted vinylpyridines were employed, the desired products (**3ll**–**rr**) were obtained in 43–89% yields. Notably, the electron-rich indole and benzofuran could survive this radical process, indicative of good chemoselectivity (**3pp** and **3qq**). Beside terminal alkenes, α,β-disubstituted alkenylpyridines successfully delivered the products (**3ss**–**uu**) in practically useful yields and with moderate diastereoselectivity. A gram-scale experiment demonstrated that this protocol could be easily scaled up (**3cc**, 5 mmol scale).

To highlight the structural diversification of ketones, other kinds of styrenes were investigated and it was found that they could uniformly react with an aromatic carboxylic acid (**1a**) to give the desired ketone products (**3vv**–**jJ**) in moderate to good yields under the optimized conditions. Its synthetic potential is distinguished by excellent and important functional group compatibility as ketone, thioether, terminal olefin/alkyne, and ester are tolerated. Due to the ubiquity of 1,1-diarylalkane scaffolds in pharmaceuticals and natural products[31], representative 1,1-diaryl olefins were used and they were found to deliver 3,3-diaryl-propanones (**3ll**–**rr** and **3cC**–**jJ**) in 43–89% yields.

The γ-carbonyl ester and γ-diketones are significant raw materials for the construction of five-membered heterocycle frameworks but their efficient and general accessibility in contemporary synthetic chemistry remains highly challenging[32,33]. As illustrated in Fig. 3 (lower part), the direct deoxygenative C–C bond formation of carboxylic acids (**1a**) with varied electron-deficient alkenes allows for modular synthesis of a diverse array of important γ-carbonyl esters, γ-carbonyl aldehydes, and γ-diketones (**3kK**–**vV**) in 51–86% yields. Its success arguably could complement the classical Weinreb ketone syntheses from Weinreb amides and Grignard reagents[34].

**Examining functional group compatibility**. After observation of the broad substrate scope, we turned our attention upon examining the functional group compatibility[35,36] with addition of a wide array of biomolecules, including natural amino acids, nucleic acids, and proteins into the reaction mixture. To avoid the use of inorganic bases, a neutral phosphate saline buffer (pH 7.4) was used. We found that the deoxygenative ketone synthesis occurred smoothly in neutral buffer-DCM solvent without any compromise of the synthetic efficiency in the presence of stoichiometric amounts of unprotected biomolecules such as L-cysteine, L-tyrosine, L-methionine, guanosine, naringin, DNA, miRNA, and bovine serum albumin (Fig. 4). In comparison with our previous deoxygenative coupling using stoichiometric

**Fig. 2** Aromatic carboxylic acid scope. Bn benzyl; Boc *tert*-butoxycarbonyl; Ts *para*-toluenesulfonyl

dimethyldicarbonates and (TMS)$_3$SiH[23], this clearly demonstrates that the deoxygenative ketone synthesis strategy has an excellent functional group compatibility. Significantly, the selective deoxygenation of only aromatic carboxylic acids in the presence of natural amino acids further underscores its synthetic advantages as amino acids are known[37] to tend to undergo photoredox decarboxylative coupling.

**Synthetic application**. The late-stage modification of complex molecules is a basis for the evaluation of a practical protocol. In this context, several biologically important natural products, pharmaceuticals, and agrochemicals were successfully used in this reaction and are shown in Fig. 5a. Three pharmaceuticals with an aromatic acid unit, telmisartan (**4**), hiestrone (**5**), and adapalene (**6**) readily underwent this deoxygenative ketone synthesis. Moreover, 11 complex alkene substrates bearing varying functional groups could be employed, affording the desired products (**7–17**) in moderate yields in aqueous solution. Interestingly, when two competing electron-deficient alkenes were assembled into one molecule, site-specific hydroacylation occurred at the less sterically hindered site (**15**). These examples clearly suggest that this strategy represents a promising late-stage application of both carboxylic acids and alkenes, and has the potential to rapidly convert two widely available starting materials into complex ketone molecules.

To underline its synthetic potential, we have applied this deoxygenative coupling protocol to synthesize the drug zolpidem, which ranks 28 in the 200 top-selling drugs (https://njardarson. lab.arizona.edu/sites/njardarson.lab.arizona.edu/files/2016Top200 PharmaceuticalPrescriptionSalesPosterLowResV2.pdf). As illustrated in Fig. 5b, six steps are usually required for the synthesis of zolpidem[38] from air-sensitive 2-bromo-1-(*p*-tolyl)ethan-1-one with 19% total yield, but we achieved a concise three-step synthesis of zolpidem with 50% total yield. First, application of deoxygenative C–C coupling of *para*-methylbenzoic acid (**1a**) with the electron-deficient alkene (**18**) successfully produced a γ-carbonyl amide (**19**) and a subsequent cyclization revealed a simple strategy for the synthesis of zolpidem from commercially abundant *para*-methylbenzoic acids.

**Intramolecular macrocyclization**. Macrocyclic ketones are used widely as fragrances, such as muskon and zibeton. To date, the macrocyclization remains a robust but highly challenging synthetic strategy, which usually calls for very low concentrations to avoid intermolecular polymerization[39]. To further demonstrate the practicality of our method, we adapted the visible-light-mediated direct deoxygenation C–C coupling to an elegant macrocyclization under optimized conditions in aqueous solution. As shown in Fig. 6,[18–20] membered cyclophane-braced cycloketones (**21a–c**) have been successfully constructed in

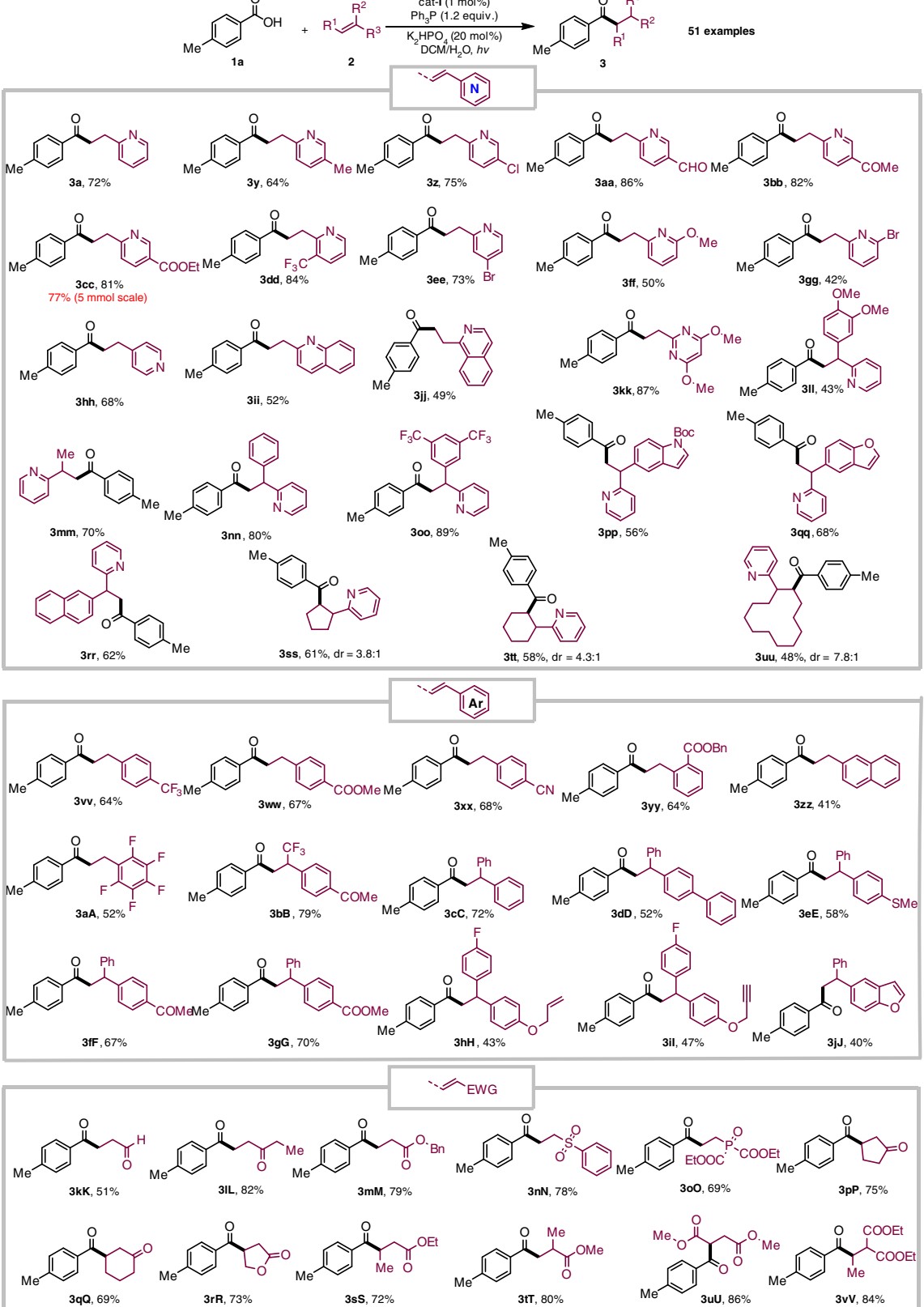

**Fig. 3** Alkene scope

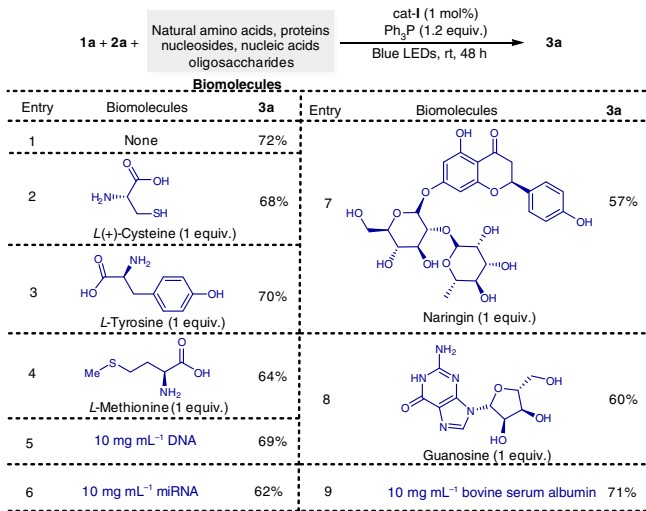

**Fig. 4** Examining functional group compatibility. Reaction conditions: photocatalyst **I** (1 mol%), **1a** (0.1 mmol, 1.0 equiv.), **2a** (0.15 mmol, 1.5 equiv.), Ph₃P (0.12 mmol, 1.2 equiv.), biomolecules, pH 7.4 PBS buffer/DCM (1:1, 2.0 mL), blue LEDs, 25 °C

synthetically useful yields, clearly demonstrating the robustness of this synthetic protocol.

**Downstream transformation**. Heterocyclic scaffolds are prevalent motifs in bioactive compounds and natural products[40]. The ketones produced in this reaction were versatile building blocks for chemical bond formation[41,42] and could undergo a broad array of downstream organic transformations to construct structurally diverse nitrogen-containing heterocycles (**22, 24, 26,** and **28**, Fig. 7). Interestingly, with 5 mol% chiral phosphoric acid **27** as catalyst, treatment of **3ii** with Hantzsch ester furnished a benzo-fused quinolizidine scaffold (**28**) with moderate enantioselectivity[43].

**Mechanistic studies**. To gain mechanistic insight into this deoxygenative C–C coupling, radical inhibitors such as 2,2,6,6-tetramethyl-1-piperidyloxy (TEMPO) and 2,6-di-*tert*-butyl-*p*-cresol were added to the model reaction system and it was found that they completely inhibited the coupling reaction. Moreover, the corresponding acyl radical was trapped by TEMPO (Fig. 8a). This indicates that an acyl radical pathway could be possible[44–47]. The intramolecular hydroacylation of 2-allylbenzoic acid (**29**) further exemplified the intermediacy of the acyl radical species (Fig. 8b) and the deuterium-labeling experiments demonstrated water was the sole proton source for this deoxygenative hydro-acylation of alkenes (Fig. 8c)[48]. The employment of ¹⁸O-labeled water and aromatic acid (**1a′**) strongly suggested that the oxygen atom of triphenylphosphine oxide would originate from carboxylic acids rather than from water (Fig. 8d). When potassium benzoate was employed in place of benzoic acid, **3d** was obtained in moderate yield, which suggests that the triphenylphosphine radical cation reacts with an aromatic carboxylate anion (Fig. 8e). The corresponding Stern–Volmer studies further revealed that photoexcited *[Ir(dF(CF₃)ppy)₂(dtbbpy)]PF₆ was quenched by triphenylphosphine (see Supplementary Information).

A possible mechanism is proposed in Fig. 8f. The photoexcited *Ir[dF(CF₃)ppy]₂(dtbbpy)]PF₆ [$E_{1/2}^{red}$ (*Ir$^{III}$/Ir$^{II}$) = +1.21 V vs SCE; $\tau$ = 2.3 μs][49] is able to undergo single-electron transfer (SET) oxidation with Ph₃P ($E_{1/2}^{red}$ = +0.98 V vs SCE)[50] to form the triphenylphosphine radical cation (**31**), which could trigger the proposed radical deoxygenation[51]. The resulting radical

cation (**31**) reacts with carboxylate anion to generate the phosphoryl radical (**32**). This is followed by β-selective C(acyl)–O bond cleavage with thermodynamic impetus for the formation of Ph₃P=O. The acyl radical (**33**) generated in this way then selectively attacks the alkene to form the radical species (**34**), which is capable of undergoing an SET with reductive Ir$^{II}$ species to afford the corresponding ketone in the presence of water. Alternatively, the homocoupling of acyl radicals (**33**) can afford a little amount of 1,2-diketones as byproducts.

**Three-component reductive coupling**. Based on this reductive quenching mechanism, we extended this deoxygenative catalytic system to an attractive three-component reductive coupling reaction of carboxylic acids (**1**), primary amines (**36**), and aromatic aldehydes (**37**). To our delight, the resulting valuable α-amino ketone products (**38**) were obtained in moderate yields (Fig. 9).

## Discussion

In summary, a deoxygenative ketone synthesis from aromatic carboxylic acids and alkenes has been developed in aqueous solution enabled by visible-light photoredox catalysis with commercially cheap triphenylphosphine as an oxygen transfer reagent. This catalytic system enables direct deoxygenation of aromatic acids to generate acyl radical in the presence of a broad variety of biomolecules. This ketone synthesis strategy allows practical and friendly reaction conditions, which significantly broadens the substrate scope, improves the functional group compatibility, and emphasizes the synthetic application in complex molecules. Based on the direct deoxygenative mechanism, a reductive three-component coupling reaction of amines, aldehydes and acids has been achieved. It offers not only a strategy for the streamlined synthesis of structurally diverse ketones from abundant carboxylic acids, but also a photoredox radical activation mode beyond the redox potential of carboxylic acids.

## Methods
**General methods**. See Supplementary Methods for further details.

**General procedure for the synthesis of 3**. To a 10 mL Schlenk tube equipped with a magnetic stir bar was added aromatic carboxylic acid **1** (0.2 mmol, 1.0 equiv.), photocatalyst Ir[dF(CF₃)ppy]₂(dtbbpy)]PF₆ (2.3 mg, 1 mol%), K₂HPO₄ (7.0 mg, 20 mol%), and Ph₃P (62.9 mg, 0.24 mmol, 1.2 equiv.) and the tube was evacuated and backfilled with argon for three times. The alkenes **2** (0.3 mmol, 1.5 equiv.) in DCM/H₂O (2.0 mL, 4:1 v/v) were added by syringe under argon. The tube was then sealed and was placed at a distance (app. 5 cm) from 5 W blue light-emitting diode (LED) lamp, and the mixture was stirred for 36–60 h at room temperature. After completion, the mixture was quenched with water and extracted with DCM (3 × 10 mL). The combined organic layer was dried over anhydrous Na₂SO₄, then the solvent was removed under vacuo. The residue was subjected to chromatography column on silica gel (eluent: petroleum ether/ethyl acetate) to give the corresponding ketone products **3**.

**General procedure for the synthesis of 4–8**. To a 10 mL Schlenk tube equipped with a magnetic stir bar was added aromatic carboxylic acid **1** (0.1 mmol, 1.0 equiv.), photocatalyst Ir[dF(CF₃)ppy]₂(dtbbpy)]PF₆ (2.3 mg, 2 mol%), K₂HPO₄ (3.5 mg, 20 mol%), and Ph₃P (31.5 mg, 0.12 mmol, 1.2 equiv.) and the tube was evacuated and backfilled with argon for three times. The alkenes **2** (0.15 mmol, 1.5 equiv.) in DCM/H₂O (2.0 mL, 4:1 v/v) were added by syringe under argon. The tube was then sealed and was placed at a distance (app. 5 cm) from 5 W blue LED lamp, and the mixture was stirred for 48 h at room temperature. After completion, the mixture was quenched with water and extracted with DCM (3 × 10 mL). The combined organic layer was dried over anhydrous Na₂SO₄, then the solvent was removed under vacuo. The residue was purified with chromatography column on silica gel (eluent: petroleum ether/ethyl acetate) to give the corresponding ketone products **4–8**.

**General procedure for the synthesis of 9–17**. To a 10 mL Schlenk tube equipped with a magnetic stir bar was added aromatic carboxylic acid **1** (0.15 mmol,

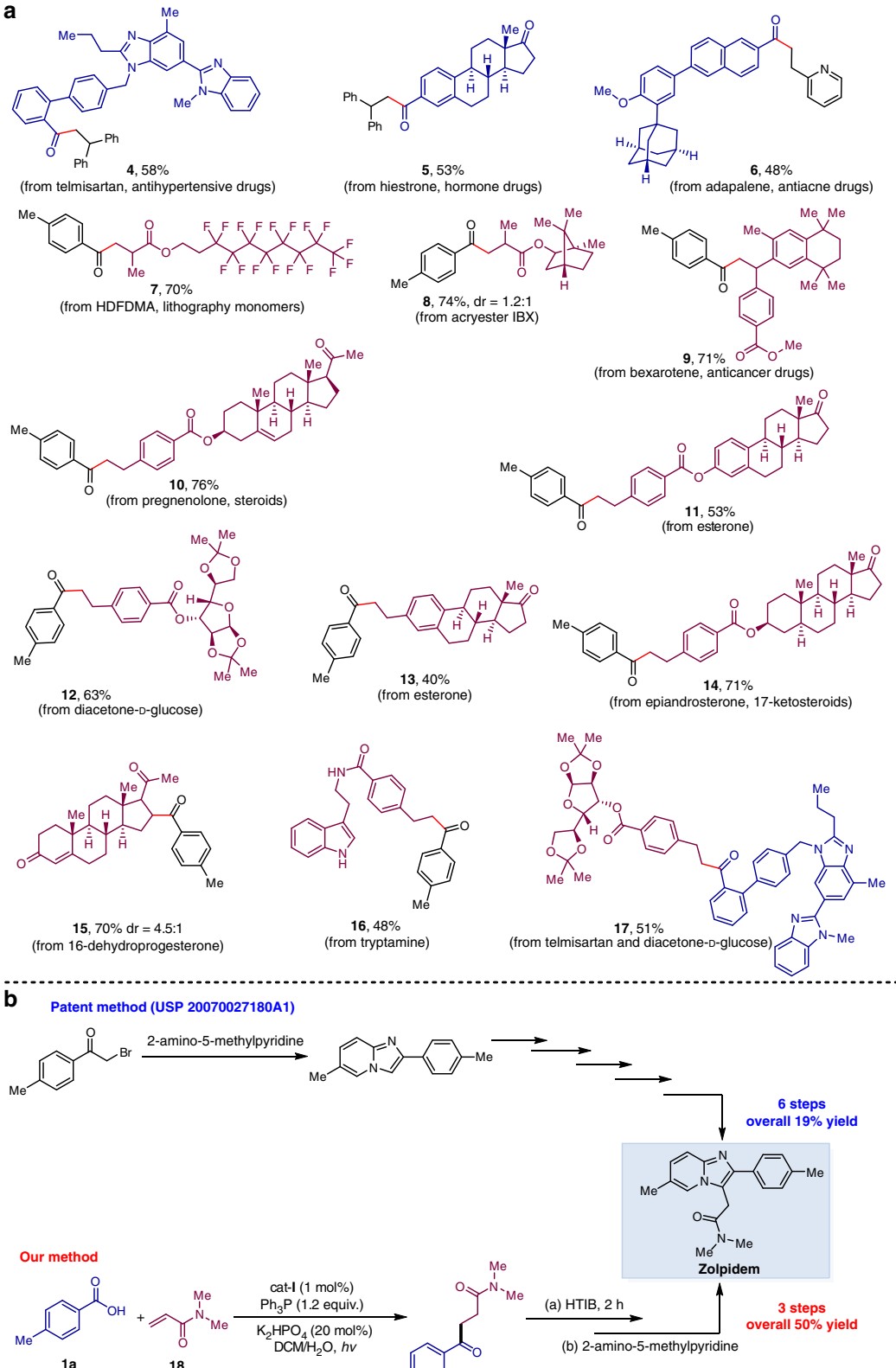

**Fig. 5** Synthetic application. **a** Late-stage application in the complex molecules. **b** Synthesis of zolpidem

1.5 equiv.), photocatalyst Ir[dF(CF₃)ppy]₂(dtbbpy)PF₆ (2.3 mg, 2 mol%), K₂HPO₄ (7.0 mg, 40 mol%), and Ph₃P (31.5 mg, 0.12 mmol, 1.2 equiv.) and the tube was evacuated and backfilled with argon for three times. The alkenes (0.1 mmol, 1.0 equiv.) in DCM/H₂O (2.0 mL, 4:1 v/v) were added by syringe under argon. The

tube was then sealed and was placed at a distance (app. 5 cm) from 5 W blue LED lamp, and the mixture was stirred for 48 h at room temperature. After completion, the mixture was quenched with water and extracted with DCM (3 × 10 mL). The combined organic layer was dried over anhydrous Na₂SO₄, then the solvent was

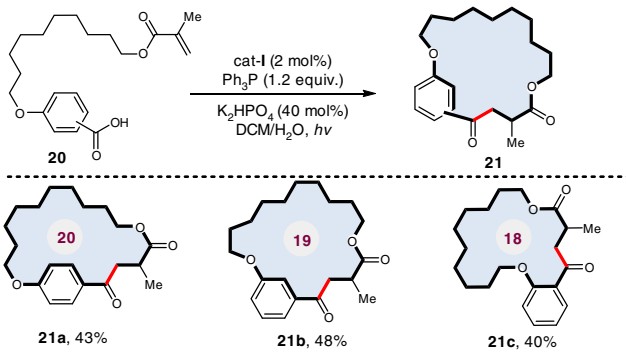

**Fig. 6** Deoxygenative macrocyclization in the synthesis of cyclophane-braced cycloketones

**Fig. 7** Downstream transformations

removed under vacuo. The residue was purified with chromatography column on silica gel (eluent: petroleum ether/ethyl acetate) to give the corresponding ketone products **9–17**.

**General procedure for the synthesis of 21.** To a 10 mL Schlenk tube equipped with a magnetic stir bar was added aromatic carboxylic acid (36.2 mg, 0.1 mmol, 1.0 equiv.), photocatalyst Ir[dF(CF$_3$)ppy]$_2$(dtbbpy)PF$_6$ (2.3 mg, 2 mol%), K$_2$HPO$_4$ (7.0 mg, 40 mol%), and Ph$_3$P (31.5 mg, 0.12 mmol, 1.2 equiv.) and the tube was evacuated and backfilled with argon for three times. Then, DCM/H$_2$O (2.0 mL, 4:1 v/v) were added by syringe under argon. The tube was then sealed and was placed at a distance (app. 5 cm) from 5 W blue LED lamp, and the mixture was stirred under room temperature for 48 h. After completion, the mixture was quenched with water and extracted with DCM (3 × 10 mL). The combined organic layer was dried over anhydrous Na$_2$SO$_4$, then the solvent was removed under vacuo. The residue was purified with chromatography column on silica gel (eluent: petroleum ether/ethyl acetate) to give the corresponding macrocyclic products **21**.

**General procedure for the synthesis of 38.** To a 10 mL Schlenk tube equipped with a magnetic stir bar was added aromatic carboxylic acid **1** (0.15 mmol, 1.5 equiv.), photocatalyst Ir[dF(CF$_3$)ppy]$_2$(dtbbpy)PF$_6$ (1.2 mg, 1 mol%), K$_2$HPO$_4$ (26.1 mg, 1.5 equiv.), and Ph$_3$P (31.5 mg, 0.12 mmol, 1.2 equiv.) and the tube was evacuated and backfilled with argon for three times. The amines **36** (0.1 mmol, 1.0 equiv.) and aldehydes **37** (0.15 mmol, 1.5 equiv.) in DCM (2.0 mL) were added by syringe under argon. The tube was then sealed and was placed at a distance (app. 5 cm) from 5 W blue LED lamp, and the mixture was stirred at room temperature for 48 h. After completion, the solvent was removed

**Fig. 8** Mechanistic studies. **a** Control experiments with additives. **b** Radical cyclization experiment. **c** Deuterium-labeling experiments. **d** $^{18}$O-labeling experiments. **e** Aromatic carboxylate anion as substrate. **f** Proposed mechanism

**Fig. 9** A three-component reductive coupling reaction

under vacuo. The resulting residue was subjected to chromatography column on silica gel (eluent: petroleum ether/ethyl acetate) to give the corresponding α-amino ketone products **38**.

## Data availability

The authors declare that all other data supporting the findings of this study are available within the article and Supplementary Information files, and also are available from the corresponding author upon reasonable request.

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

## Acknowledgements

We gratefully acknowledge the National Natural Science Foundation of China (21702098, 21672099, and 21732003), "1000 Youth Talents Plan", and the Fundamental Research Funds for the Central Universities (No. 020514380158 and 020514380131).

M.Z. was supported by Nanjing University Innovation and Creative Program for Ph.D. candidate (No. CXCY17-19). Prof. Carl Johan Wallentin at Gothenburg University (Sweden) is kindly acknowledged for his helpful paper discussion.

## Author contributions

M.Z. performed and analyzed the experiments. M.Z., J.X. and C.Z. co-wrote and discussed the manuscript. J.X. conceived and directed the whole project. All authors commented on the final manuscript and contributed to the analysis and interpretation of the results.

## Additional information

**Competing interests:** The authors declare no competing interests.

