## [Peer Review File · Nature Communications]

Reviewer #1 (Remarks to the Author):

The communication by Zhu, Xie and coworkers describes an acyl radical addition to alkenes with aromatic carboxylic acids as acyl precursors by photoredox catalysis. The generation of acyl radicals from aromatic carboxylic acids and stoichiometric amount of triphenylphosphine is unknown, however the concept is quite similar to previously reported mixed anhydride method for acyl radical generation from carboxylic acids. To this end, the authors have demonstrated various synthetic applications to showcase the advantage of the described method, such as the broad range of heterocycles, biomolecule-compatibility, late-stage modification, macrocyclization, and three-component coupling reaction. There are mechanistic novelties and synthetic advantages arising from the use of triphenylphosphines, which may be suitable for the publication in Nat. Commun. However, the following points and concerns need to be addressed.

- A. The yields are typically in the range of modest 50-70%, what are the side products and the related mechanistic pathways?
- B. With the generation of stoichiometric triphenylphosphine oxides precipitates, it is difficult to imagine the listed biomolecules are not affected. The authors should carefully modify the claim and conduct further experiments to test the intactness of the biomolecules.
- C. Other representative examples to generate acyl radicals should be cited.
- D. The authors should refrain from the argument of atom economical, as the triphenylphosphine is arguably one of the least atom-economical reagents in organic synthesis, and the resulting triphenylphosphine oxides are very difficult to remove.

Reviewer #2 (Remarks to the Author):

This manuscript submitted by Zhang, Xie, and Zhu describes an interesting deoxygenative ketone synthesis from carboxylic acids and alkenes. This transformation proceeds under visible-light photoredox catalyzed condition with triphenylphosphine as an oxygen transfer reagent, exhibiting a very broad substrate scope and excellent functional group tolerance with respect to both carboxylic acid and alkene components. The advantages of this protocol are further illustrated by successful application to the late-stage synthesis of complex molecules, biomolecule-compatible ketone synthesis, intramolecular macrocyclization, and a three-component reductive coupling reaction. Mechanistic experiments indicate that a key acyl radical can be generated by homolytic C(acyl)-O cleavage of carboxylic acids, and water acts as a proton source. Based on the above considerations, I

would like to recommend publication in Nature Communications after the following issues have been addressed:

1. In Scheme 1a, it should be "kcal/mol", not "kcal/mol".
2. If the authors perform the reaction with ^{18}O -labeled benzoic acid, the isolation of $\text{Ph}_3\text{P}=\text{C}^{18}\text{O}$ can better prove the proposed mechanism.
3. Does the reaction work with vinyl and aliphatic carboxylic acids?
4. Scheme 5, eq. 4: a citation to previous work (Synlett 2011, 1243-1246) should be given.
5. Did the authors try the reaction with other photocatalysts? Can organic dyes be applied?

Dear Dr. Bottari,

Thank you very much for your positive decision. We also would like to express our heartfelt gratitude to both reviewers for their helpful suggestions and positive comments on our manuscript, which can significantly improve the quality of this manuscript. According to the editorial office manuscript checklist and reviewers' comments, we carefully revised the manuscript. Please check the files.

Best regards

Prof. Dr. Jin Xie

Nanjing University

Nanjing 210023

Reviewers' comments:

Reviewer #1 (Remarks to the Author):

The communication by Zhu, Xie and coworkers describes an acyl radical addition to alkenes with aromatic carboxylic acids as acyl precursors by photoredox catalysis. The generation of acyl radicals from aromatic carboxylic acids and stoichiometric amount of triphenylphosphine is unknown, however the concept is quite similar to previously reported mixed anhydride method for acyl radical generation from carboxylic acids. To this end, the authors have demonstrated various synthetic applications to showcase the advantage of the described method, such as the broad range of heterocycles, biomolecule-compatibility, late-stage modification, macrocyclization, and three-component coupling reaction. There are mechanistic novelties and synthetic advantages arising from the use of triphenylphosphines, which may be suitable for the publication in *Nat. Commun.* However, the following points and concerns need to be addressed.

- A. The yields are typically in the range of modest 50-70%, what are the side products and the related mechanistic pathways?
- B. With the generation of stoichiometric triphenylphosphine oxides precipitates, it is difficult to imagine the listed biomolecules are not affected. The authors should carefully modify the claim and conduct further experiments to test the intactness of the biomolecules.
- C. Other representative examples to generate acyl radicals should be cited.
- D. The authors should refrain from the argument of atom economical, as the triphenylphosphine is arguably one of the least atom-economical reagents in organic synthesis, and the resulting triphenylphosphine oxides are very difficult to remove.

Reviewer #2 (Remarks to the Author):

This manuscript submitted by Zhang, Xie, and Zhu describes an interesting deoxygenative ketone synthesis from carboxylic acids and alkenes. This transformation proceeds under visible-light photoredox catalyzed condition with triphenylphosphine as an oxygen transfer reagent, exhibiting a very broad substrate scope and excellent functional group tolerance with respect to both carboxylic acid and alkene components. The advantages of this protocol are further illustrated by successful application to the late-stage synthesis of complex molecules, biomolecule-compatible ketone synthesis, intramolecular macrocyclization, and a three-component reductive coupling

reaction. Mechanistic experiments indicate that a key acyl radical can be generated by hemolytic C(acyl)-O cleavage of carboxylic acids, and water acts as a proton source. Based on the above considerations, I would like to recommend publication in Nature Communications after the following issues have been addressed:

1. In Scheme 1a, it should be “kcal/mol”, not “kacl/mol”.
2. If the authors perform the reaction with ^{18}O -labeled benzoic acid, the isolation of $\text{Ph}_3\text{P}=\text{O}$ can better prove the proposed mechanism.
3. Does the reaction work with vinyl and aliphatic carboxylic acids?
4. Scheme 5, eq. 4: a citation to previous work (Synlett 2011, 1243-1246) should be given.
5. Did the authors try the reaction with other photocatalysts? Can organic dyes be applied?

Our responses

Reviewer 1

A. The yields are typically in the range of modest 50-70%, what are the side products and the related mechanistic pathways?

Answer: Thanks for your comment and suggestion. The acyl radicals generated from carboxylic acids may undergo homocouplings, and a little amount of 1,2-diketones were detected by GC-MS as the side products.

The t_R and MS-EI match well with the known 1,2-diketone references.

According to your comments, we added the side reaction of homocoupling in the mechanistic proposal (Fig. 8f) as the following:

One new sentence was added in the manuscript as “Alternatively, the homocoupling of acyl radicals (33) can afford a little amount of 1,2-diketones as byproducts.” In addition, these new results were added into the Supplementary Information.

B. With the generation of stoichiometric triphenylphosphine oxides precipitates, it is difficult to imagine the listed biomolecules are not affected. The authors should carefully modify the claim and conduct further experiments to test the intactness of the biomolecules.

Answer: Thanks for your helpful suggestion. We strongly agree with you. In the presence of stoichiometric amounts of biomolecules, the deoxygenation reaction proceeds smoothly and is not significantly influenced. After careful considerations, we also think direct “biomolecule-compatible” description is not very appropriate in this manuscript. According to your suggestion, we changed “examining biomolecule-compatible” to “examining functional group compatibility”. In the revised manuscript, we have deleted “biomolecule-compatible” from title and text. For example, in the manuscript, we also have changed “biocompatibility” to “functional group compatibility”. One sentence was used in the revised manuscript as “After observation of the broad substrate scope, we turned our attention upon examining the functional group compatibility^{35,36} with addition of a wide array of biomolecules including natural amino acids, nucleic acids and proteins into the reaction mixture.”

In addition, according to your comments, we conducted some experiments to test the intactness of added biomolecules. For example, after the reaction finished, amino acid molecules can be converted to salts by adding 1M HCl aq. to the reaction mixture. These biomolecules can be proved by ¹H NMR and MS-ESI. From MS-ESI (negative mode), the added amino acid molecules can be found and its formation was analyzed by ¹H NMR analysis. This implies that added biomolecule remains intact in the reaction mixture. These new results were added in the revised Supplementary Information.

Line# 1 R. Time:----(Scan#----)
 MassPeaks:82
 Spectrum Mode:Averaged 0.150-0.283(10-18) Base Peak:184(46450)
 BG Mode:Averaged 0.083-0.483(6-30) Segment 1 - Event 2

Line#: 1 R Time: --- (Scan#: ---) MS Spectrum
 MassPeaks: 112
 Spectrum Mode: Averaged 0.117-0.283(8-18) Base Peak: 216(6381)
 BG Mode: Averaged 0.050-0.550(4-34) Segment 1 - Event 2

In addition, naringin and guanosine can also be found from MS-ESI after the reaction finished.

MS-ESI (negative): [M-1]

Line#: 2 R Time: ----(Scan#: ----)
MassPeaks: 86
Spectrum Mode: Averaged 0.150-0.250(10-16) Base Peak: 282(6390)
BG Mode: Averaged 0.083-0.450(6-28) Segment 1 - Event 2

Line#: 2 R Time: ----(Scan#: ----)
MassPeaks: 200
Spectrum Mode: Averaged 0.150-0.283(10-18) Base Peak: 579(45828)
BG Mode: Averaged 0.083-0.550(6-34) Segment 1 - Event 2

MS-ESI (negative): [M-1]

C. Other representative examples to generate acyl radicals should be cited.

Answer: Thanks for your suggestion, and the recent representative references about acyl radicals have been cited in the revised manuscript (see references 45-48).

45. Liu, J. *et al.* Visible-light-mediated decarboxylation/oxidative amidation of α -keto acids with amines under mild reaction conditions using O_2 . *Angew. Chem. Int. Ed.* 53, 502-506 (2014).

46. Mukherjee, S., Garza-Sanchez, R. A., Tlahuext-Aca, A. & Glorius, F. Alkynylation of C(O)-H bonds enabled by photoredox-mediated hydrogen-atom transfer. *Angew. Chem. Int. Ed.* 56, 14723-14726 (2017).

47. Jia, K., Pan, Y. & Chen, Y. Selective carbonyl-C(sp³) bond cleavage to construct ynamides, ynoates, and ynones by photoredox catalysis. *Angew. Chem. Int. Ed.* 56, 2478-2481 (2017).

48. Zhang, X. & MacMillan, D. W. C. Direct aldehyde C-H arylation and alkylation via the combination of nickel, hydrogen atom transfer, and photoredox catalysis. *J. Am. Chem. Soc.*

139, 11353-11356 (2017).

D. The authors should refrain from the argument of atom economical, as the triphenylphosphine is arguably one of the least atom-economical reagents in organic synthesis, and the resulting triphenylphosphine oxides are very difficult to remove.

Answer: Thank you very much for your kind suggestion. According to your comments, we have modified the statement and removed the argument of “atom economical” from the manuscript.

Reviewer 2

1. In Scheme 1a, it should be “kcal/mol”, not “kcal/mol”.

Answer: Thank you for pointing out our mistake. According to the manuscript checklist, “kcal mol⁻¹” was used.

2. If the authors perform the reaction with ¹⁸O-labeled benzoic acid, the isolation of Ph₃P=¹⁸O can better prove the proposed mechanism.

Answer: Thank you very much for your helpful suggestion. When ¹⁸O-labeled 4-methylbenzoic acid (93% ¹⁸O incorporation) was subjected to our standard conditions, ¹⁸O-labeled triphenylphosphine oxides (40% ¹⁸O incorporation) and ¹⁸O-labeled ketones (44% ¹⁸O incorporation) were obtained respectively, which suggested the proposed pathway was likely. The result has been added in the revised manuscript (See Fig. 9d in the revised manuscript) and also Supplementary Information. One new sentence was added in the manuscript as “The employment of ¹⁸O-labeled water and aromatic acid (1a') strongly suggested that the oxygen atom of triphenylphosphine oxide would originate from carboxylic acids rather than from water (Eq 10, 11).”

3. Does the reaction work with vinyl and aliphatic carboxylic acids?

Answer: Thanks very much for your helpful suggestion. According to your comments, aliphatic carboxylic acids such as primary, secondary and tertiary alkyl carboxylic acids were subjected to standard conditions, and no desired products were observed. The α,β -unsaturated carboxylic acids and 3-phenylpropionic acid also did not give the ketone

products. We have added these unsuccessful experiments in Fig. 2 in the revised manuscript to let the readers know its current limitation. One new sentence was added in the revised manuscript as “Besides aromatic acids, other kinds of carboxylic acids including aliphatic carboxylic acids and α,β -unsaturated carboxylic acids however failed in this reaction.”

4. Scheme 5, eq. 4: a citation to previous work (Synlett 2011, 1243-1246) should be given.

Answer: Thanks. The reference has been cited in the revised manuscript as reference 44.

44. Rueping, M. & Hubener, L. Enantioselective synthesis of quinolizidines and indolizidines via a catalytic asymmetric hydrogenation cascade. *Synlett* 1243-1246 (2011).

5. Did the authors try the reaction with other photocatalysts? Can organic dyes be applied?

Answer: Thanks for your suggestion. Other photocatalysts such as Ru-, Ir-based photocatalysts and organic dyes proved ineffective. The new results have been added in Table 1 in revised manuscript.

Table 1. Optimization of reaction conditions

Entry	Variation of conditions	Yield ^a
1	none	72%
2	without H ₂ O	40%
3	no Ph ₃ P	nd
4	1 mol% cat-II instead of cat-I	nd
5	2 mol% cat-III instead of cat-I	nd
6	2 mol% cat-IV instead of cat-I	nd
7	2 mol% cat-V instead of cat-I	nd
8	no photocatalyst	nd
9	no light	nd

Standard conditions: photocatalyst **I** (1 mol%), **1a** (0.2 mmol), **2a** (1.5 equiv), Ph₃P (1.2 equiv), K₂HPO₄ (20 mol%), DCM/H₂O (2.0 mL, v/v = 4:1), rt, 5 W blue LEDs, 48 h. ^a Isolated yield.

One new sentence was added in the revised manuscript as “Other photocatalysts such as *fac*-Ir(ppy)₃ (II), Ru(bpy)₃(PF₆)₂ (III), Eosin Y (IV) and Acr⁺-Mes (V) were proved ineffective for this transformation (Table 1, entries 4-7).”

Reviewer #1 (Remarks to the Author):

The authors have made the changes accordingly.

Reviewer #2 (Remarks to the Author):

The authors have adressed all mentioned issues in full and have improved the work.

I strongly support publication of this fine manuscript.

Response to the referees

REVIEWERS' COMMENTS:

Reviewer #1 (Remarks to the Author):

The authors have made the changes accordingly.

Response: Thank you very much.

Reviewer #2 (Remarks to the Author):

The authors have adressed all mentioned issues in full and have improved the work.
I strongly support publication of this fine manuscript.

Response: Thank you very much.